# Kaposi's sarcoma-associated herpesvirus latency-associated nuclear antigen dysregulates expression of MCL-1 by targeting FBW7

Yeong Jun Kim[1], Yuri Kim[2], Abhishek Kumar[3], Chan Woo Kim[1], Zsolt Toth[3], Nam Hyuk Cho[2,4], Hye-Ra Lee[1,5]*

**1** Department of Biotechnology and Bioinformatics, College of Science and Technology, Korea University, Sejong, South Korea, **2** Department of Microbiology and Immunology, Seoul National University college of Medicine, Seoul, South Korea, **3** Department of Oral Biology, University of Florida College of Dentistry, Gainesville, Florida, United States of America, **4** Department of Biomedical Sciences, Seoul National University college of Medicine, Seoul, South Korea, **5** Department of Lab Medicine, College of Medicine, Korea University, Seoul, South Korea

* leehr@korea.ac.kr

**Data Availability Statement:** All relevant data are within the manuscript and its Supporting Information files.

## Abstract

Primary effusion lymphoma (PEL) is an aggressive B cell lymphoma that is etiologically linked to Kaposi's sarcoma-associated herpesvirus (KSHV). Despite standard multi-chemotherapy treatment, PEL continues to cause high mortality. Thus, new strategies to control PEL are needed urgently. Here, we show that a phosphodegron motif within the KSHV protein, latency-associated nuclear antigen (LANA), specifically interacts with E3 ubiquitin ligase FBW7, thereby competitively inhibiting the binding of the anti-apoptotic protein MCL-1 to FBW7. Consequently, LANA-FBW7 interaction enhances the stability of MCL-1 by preventing its proteasome-mediated degradation, which inhibits caspase-3-mediated apoptosis in PEL cells. Importantly, MCL-1 inhibitors markedly suppress colony formation on soft agar and tumor growth of KSHV⁺PEL/BCBL-1 in a xenograft mouse model. These results strongly support the conclusion that high levels of MCL-1 expression enable the oncogenesis of PEL cells and thus, MCL-1 could be a potential drug target for KSHV-associated PEL. This work also unravels a mechanism by which an oncogenic virus perturbs a key component of the ubiquitination pathway to induce tumorigenesis.

## Author summary

Primary effusion lymphoma (PEL), a highly aggressive B cell lymphoma, is associated with Kaposi's sarcoma-associated herpesvirus (KSHV). However, the underlying mechanisms that govern the aggressiveness of KSHV-associated PEL are poorly understood. Here, we demonstrate that KSHV LANA interacts with cellular ubiquitin E3 ligase FBW7, sequestering MCL-1 from FBW7, which reduces MCL-1 ubiquitination. As such, LANA potently stabilizes and increases MCL-1 protein, leading to inhibition of caspase-3-

**Funding:** This research was supported by the Basic Science Research Program through the National Research Foundation of Korea (NRF) (2019R1A2C1008512 to H.-R.L.; 2017M3A9E4061998 to N.H.C)(https://www.nrf.re.kr). ZT was supported by the National Institute of Health (R01AI132554) (https://www.nih.gov). The funders had not role in study design, data collection and analysis, decision to publish, or preparation of the manuscript.

**Competing interests:** The authors have declared that no competing interests exist.

mediated apoptosis in PEL cells. Furthermore, MCL-1 inhibitors efficiently blocked PEL progression in mouse xenograft model. These results suggest that LANA acts as a proto-oncogene via deregulating tumor suppressor FBW7, which upregulates anti-apoptotic MCL-1 expression. This study suggests drugs that target MCL-1 may serve as an effective therapy against KSHV[+] PEL.

## Introduction

The ubiquitin-proteasome system (UPS) is a network of proteins involved in ubiquitination of cellular targets and subsequent control of numerous cellular functions. Dysregulation of the components of this elaborate network leads to human diseases such as cancer [1]. One pre-dominantly dysregulated component of UPS is an E3 ubiquitin ligase called F-box and WD repeat domain-containing 7 (FBW7) that specifically recognizes phosphorylated substrates within the conserved Cdc4 phosphodegron (CPD) motif [2,3]. Notably, FBW7 has been classi-fied as a tumor suppressor that induces degradation of several proto-oncogenes such as c-Myc, cyclin E, and myeloid cell leukemia-1 (MCL-1) [2–4]. Among them, MCL-1, which belongs to the anti-apoptotic BCL-2 family, prevents mitochondrial outer membrane permeabilization by inactivating Bak and Bax, thus blocking caspase-3-mediated apoptosis [5,6]. Moreover, MCL-1 is frequently amplified or overexpressed in a broad spectrum of cancers to promote tumor cell survival, suggesting its oncogenic role [7,8]. Interestingly, MCL-1 is highly expressed in tis-sues from patients with primary effusion lymphoma (PEL) [9–11].

PEL, caused by Kaposi's sarcoma-associated herpesvirus (KSHV), is an aggressive B cell malignancy that occurs in peritoneal spaces and body cavities (e.g. pleural and pericardial) [12–14]. The prognosis is usually poor, with a median survival of 6.2 months [15]. More importantly, there are still no efficient therapeutic agents for PEL. The PEL cells are predomi-nantly infected with latent forms of KSHV, while only a small subset of PEL cells undergoes spontaneous lytic replication [13,16]. Like other herpesviruses, KSHV displays two distinct stages of its life cycle: latent and lytic. Latent KSHV expresses a limited number of genes, which regulates PEL cell proliferation, apoptosis, and pathogenesis [17,18]. Among them, latency-associated nuclear antigen (LANA) is a key latent gene that serves as an essential multi-functional viral protein. First, LANA tethers viral episomal DNA to mitotic host chromosomes to allow segregation of episomes to daughter nuclei for maintaining KSHV latency [19–21]. Second, LANA promotes tumorigenic properties by either suppressing apoptotic pathways or activating oncogenic pathways through interaction with numerous cellular factors [13,16,22]. These tumorigenic properties might be related, at least in part, to the dysregulation of compo-nents of UPS to induce KSHV-associated malignancy. Indeed, several studies show that KSHV-encoding proteins hijack the cellular UPS to dysregulate expression of multiple cellular substrates, ultimately promoting successful viral infection [23–25]. However, few have assessed how KSHV dysregulates and exploits UPS to promote KSHV pathogenesis.

Here, we identified the mechanism by which KSHV drives high expression of MCL-1 by targeting E3 ubiquitin ligase FBW7 and clarified its role in KSHV-associated lymphoma. We found that KSHV LANA interacts with FBW7 in a phosphodegron motif-dependent manner and that this interaction specifically suppresses MCL-1 ubiquitination, eventually inhibiting caspase-3-dependent apoptosis. Furthermore, depletion of MCL-1 by small interfering RNA (siRNA) or pharmacological inhibition of MCL-1 using the small-molecule inhibitor AT-101 [26] markedly suppressed proliferation and colony formation on soft agar by several PEL cell lines. Remarkably, AT-101 effectively inhibited tumor development and growth in PEL-

derived mouse xenograft model. Taken together, these findings shed new light on the crucial role of MCL-1 in PEL transformation and identify MCL-1 as a potential therapeutic target for KSHV-associated PEL.

## Results

### KSHV LANA interacts specifically with FBW7 in a phosphorylation-dependent manner

Normally, FBW7 substrates harbor the CPD motif that contains a threonine or serine residue for phosphorylation in the "0" position and serine or glutamic acid residue in the "+4" position, in brief T/S-X-X-X-S/E [4]. While investigating the function of LANA, we identified two consensus CPD motifs in its N-terminal region (**Fig 1A**) that appear to be recognized by FBW7. To validate whether LANA interacts with FBW7 through its CPD motifs, we first carried out immunoprecipitation (IP) experiments to examine the interaction between LANA and FBW7. Notably, we found that LANA interacted with FBW7 (**Fig 1B**) but not with other F-box family proteins (**Fig 1C**). Since substrate phosphorylation is critical for being recognized by FBW7 [2–4], we examined whether LANA-FBW7 interaction occurs in a phosphorylation-dependent manner. The results showed that LANA dephosphorylated by λ-phosphatase no longer interacted with FBW7 (**Fig 1D**), suggesting that phosphorylation of LANA is necessary for interaction with FBW7.

To further identify the significant phosphorylation sites that are responsible for LANA-FBW7 interaction, we performed IP assays after generating three different phosphorylation-dead LANA mutants: LANA$^{T177A}$ (designated as LANA-P1), LANA$^{S219A/S223A}$ (designated as LANA-P2/P3), and LANA$^{T117A/S219A/S223A}$ (designated as LANA-P1/P2/P3). Only LANA-P2/P3 bound efficiently to FBW7, suggesting that phosphorylation at threonine 117 (P1) is critical for the LANA-FBW7 interaction (**Fig 1E**). To eliminate the possibility that different intracellular localization of LANA-P1 from LANA-WT could affect the FBW7-LANA-P1 interaction, we performed immunofluorescence assays and found that LANA-P1, similar to LANA, was also localized in the nucleus as was FBW7 (**Figs 1F** and S1). Taken together, these results suggest that phosphorylation of the first CPD motif within LANA is essential for its interaction with FBW7.

### KSHV LANA-FBW7 interaction alters expression of MCL-1 but not LANA

Given that phosphorylation of substrate is required for FBW7 to recognize and induce substrate ubiquitination [7], we investigated whether LANA-FBW7 interaction affects the stability of LANA. Cells were transfected with FBW7 along with LANA or LANA-P1, followed by exposure to cycloheximide (CHX) to block newly synthesized proteins. Intriguingly, the interaction of LANA with FBW7 did not downregulate LANA expression (**Fig 2A**), indicating that LANA is not a substrate of FBW7, even though FBW7 binds phosphorylated LANA.

Mounting evidence suggests that KSHV LANA deregulates expression of several FBW7 substrates, including c-Myc, Notch-1, and cyclin E, in BCBL-1 cells infected with KSHV [27–29]. Therefore, we speculated that LANA-FBW7 interaction potentially induces the stabilization of one of known FBW7 substrates via targeting. To examine this, we generated tetracycline-inducible TREx/BJAB cell lines that ectopically expressed Au-tagged LANA or LANA-P1. These cells were treated with doxycycline (Doxy) together with CHX for various periods of time and then harvested for immunoblot analysis with designated antibodies. Under these conditions, the half-life of MCL-1 was much longer in the presence of LANA-WT than in the presence of the empty vector or LANA-P1 (**Fig 2B**). In contrast, both LANA-WT

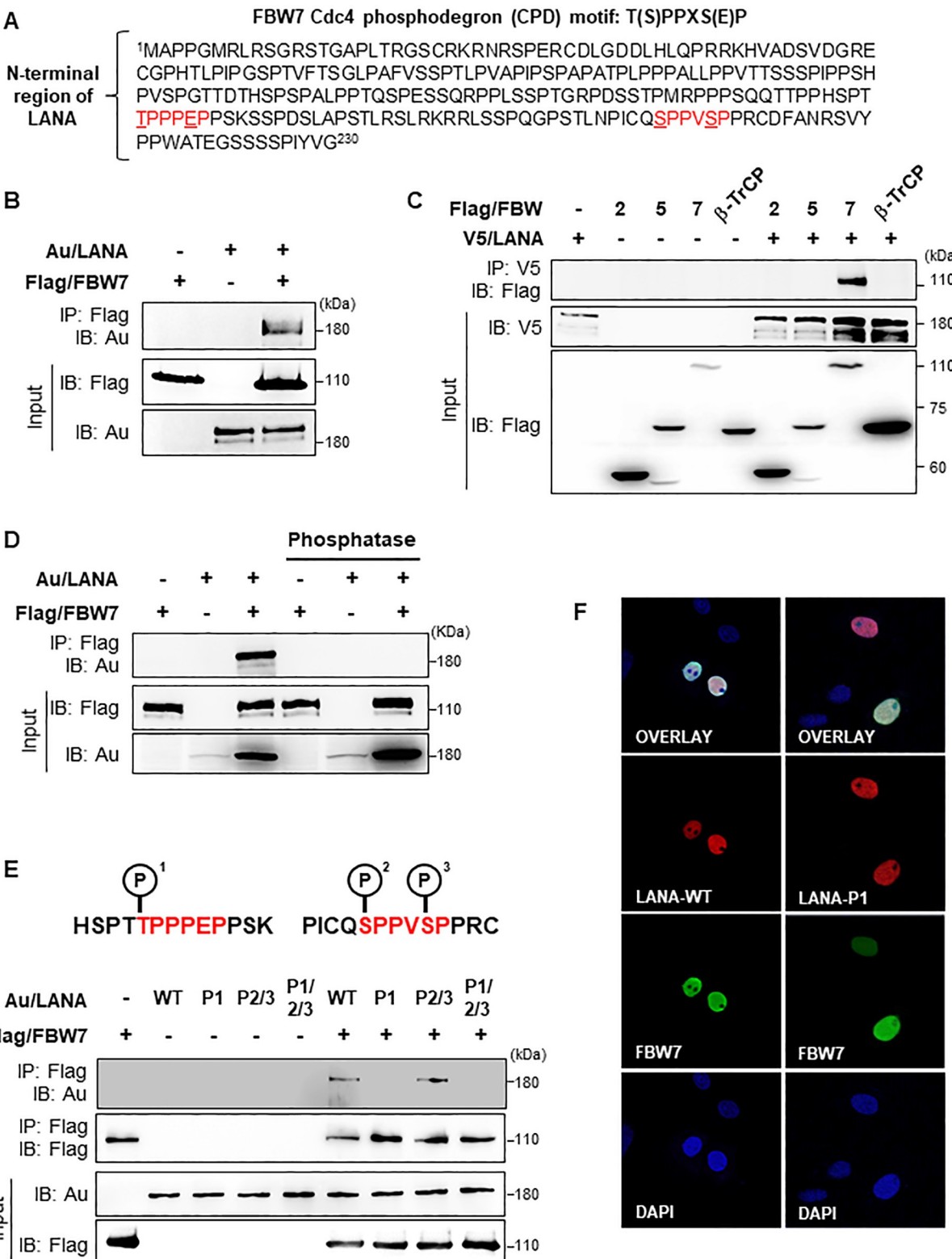

**Fig 1. KSHV LANA interacts with FBW7 in a phosphorylation-dependent manner.** (A) The red sequences represent the FBW7 consensus binding motif present in the N-terminal region of LANA. (B) 293T cells were transfected with the indicated constructs, followed by immunoprecipitation (IP) with an anti-Flag antibody and immunoblotting (IB) with an anti-Au antibody. (C) 293T cells were co-transfected with LANA and the indicated F-box/WD repeat constructs, followed by IP with an anti-V5 antibody and IB with an anti-Flag antibody. (D)

293T cells were co-transfected with Au/LANA and Flag/FBW7. Cell lysates were treated with or without phosphatase, followed by IP with an anti-Flag antibody and IB with an anti-Au antibody. (E) 293T cells were transfected with the indicated combinations of phosphorylation-dead mutant plasmids followed by IP and IB. (F) Vero cells transiently expressing wild-type (WT) LANA or LANA-P1 together with Flag/FBW7, respectively, were fixed, stained with an anti-Au antibody (red) and an anti-Flag antibody (green), and viewed under a confocal microscope. Nuclei (blue) were stained with DAPI.

and LANA-P1 elevated the intracellular domain of Notch-1 (ICN) expression (**Fig 2B**). We also found that LANA expression did not change the protein levels of other FBW7's substrates such as mTOR and cyclin E indicating that LANA-FBW7 interaction specifically alters MCL-1 expression (**Fig 2B**). We note that a previous study showed that the interaction of the C-terminal region of LANA with SEL10 (also known as FBW7) induced the stabilization of ICN [27]. In consistent with this study [27], our results showed that both LANA and LANA-P1 accumulated ICN, while LANA, but not LANA-P1, stabilized MCL-1 (**Fig 2B**). Our data suggested that MCL-1 is stabilized primarily through the binding of phosphorylated LANA to FBW7. Further analysis of the mechanism by which LANA increases the stability of MCL-1 via interaction with FBW7 revealed that MCL-1 was degraded rapidly when co-expressed with FBW7 (**Fig 2C**, compared lanes 1–4 with lanes 5–8). However, the half-life of MCL-1 was greatly increased in the presence of LANA but not in the presence of LANA-P1 (**Fig 2C**, lanes 9–12, compared the upper and lower panels). Collectively, these results indicate that LANA-FBW7 interaction stabilizes MCL-1.

## KSHV LANA stabilizes MCL-1 via competitive interaction with FBW7, leading to reduced ubiquitination of MCL-1

Since the stability of MCL-1 protein is determined primarily by the interaction with E3 ubiquitin ligase FBW7 through phosphodegron sequences of MCL-1 [7,30], we hypothesized that LANA might compete with MCL-1 for binding to FBW7, thereby increasing MCL-1 protein levels by blocking its ubiquitin-proteasome-dependent degradation. To address this, we performed competitive protein-binding assays with cells transiently expressing different combinations of LANA-WT, LANA-P1, MCL-1, and FBW7 (**Fig 3A**). The results showed that as LANA levels increased, FBW7 binding to MCL-1 prominently decreased, but the interaction of FBW7 with MCL-1 was not affected by expression of LANA-P1 (**Fig 3A**, lanes 8–11). Next, we investigated whether sequestration of FBW7 by LANA affected FBW7-mediated ubiquitination of MCL-1. Consistent with a previous report [7], we found that ubiquitination of MCL-1 absolutely relies on the presence of FBW7 (**Fig 3B** upper panel, lane 4). Importantly, LANA-WT dramatically decreased MCL-1 ubiquitination (**Fig 3B** upper panel, compared lane 4 with lane 5), while LANA-P1 marginally reduced MCL-1 ubiquitination (**Fig 3B** upper panel, compared lane 4 with lane 6). Using K48R (Lysine 48 to Arginine) ubiquitin mutant, MCL-1 polyubiquitination was marginally increased when FBW7 was simultaneously overexpressed (**Fig 3B** lower panel, compared lane 3 with lane 4) and there was no effect by either LANA or LANA-P1 expression (**Fig 3B** lower panel, compared lane 4 with lanes 5 & 6). This result indicates that MCL-1 was mainly modified by K48-linked ubiquitin chain as shown by previous report [31], and LANA suppresses K48-linked ubiquitination of MCL-1, ultimately leading to the stabilization of MCL-1 (**Fig 3B**).

MCL-1 is considered as a mitochondrial protein [5], while LANA and FBW7 are localized in the nucleus [13,32]. To further understand how nuclear proteins LANA and FBW7 regulate mitochondrial protein MCL-1, their subcellular localization in mitochondria and nucleus was explored using tetracycline-inducible TREx/BJAB cell lines that ectopically expressed Au-tagged LANA or LANA-P1 (**Fig 3C**). MCL-1 was predominantly in the mitochondria as

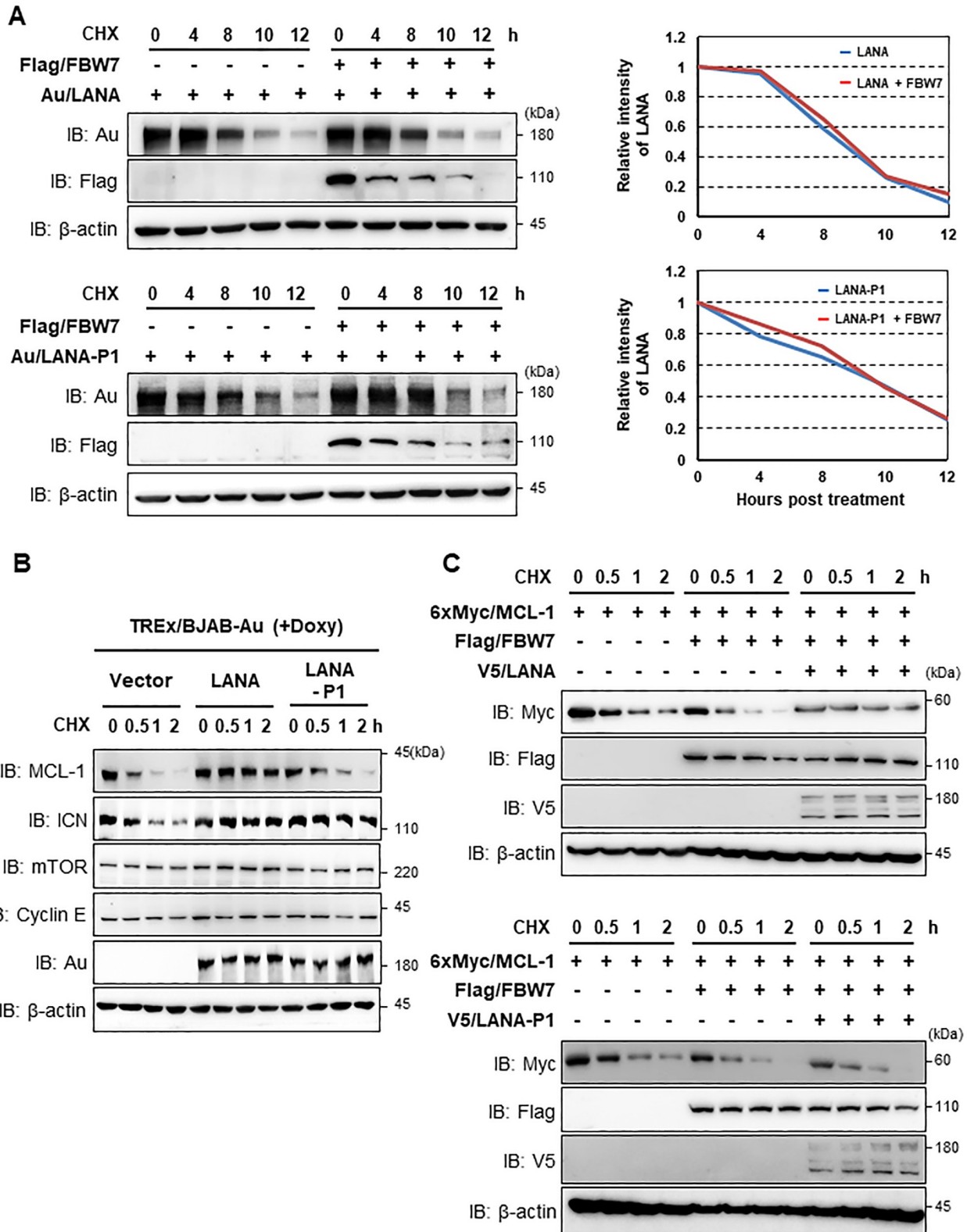

**Fig 2. The LANA-FBW7 interaction specifically stabilizes the MCL-1 protein.** (A, upper left) After transfection with LANA-WT and/or FBW7, cells were treated with cycloheximide (CHX, 20 μg/ml) for the indicated periods, followed by IB with anti-Au, anti-Flag, or anti-β-actin antibodies. (A, bottom left) At 40 h post-transfection with LANA-P1 and/or FBW7, 293T cells were treated with CHX (20 μg/ml) for the indicated periods,

followed by IB with the indicated antibodies. (A, upper and bottom right) Time course of expression was determined by semi-quantification of IB bands. (B) TREx/BJAB LANA/Au and TREx/BJAB LANA-P1/Au cells were treated with doxycycline (Doxy, 1 μg/ml) together with CHX (20 μg/ml) for the indicated periods. Equal amounts of total protein were analyzed by IB with the indicated antibodies. (C) Cells were transfected with the MCL-1 constructs together with the FBW7 and LANA or LANA-P1 constructs. Cells were then treated with CHX (20 μg/ml) for the indicated periods, followed by IB with the designated antibodies.

expected (**Fig 3C**). Interestingly, we first found that expression of LANA in TREx/BJAB cells significantly increased the nuclear localization of FBW7 compared to vector or LANA-P1 expression (**Fig 3C** lanes 5–7). More importantly, LANA expression reduced the mitochondria localization of FBW7 (**Fig 3C**, compared lanes 1 and 3 with lane 2), while highly elevated the expression of MCL-1 (**Fig 3C**, compared lanes 1 and 3 with lane 2). Consequently, these data indicate that LANA effectively sequesters FBW7 into the nucleus and impairs the FBW7-mediated ubiquitination of MCL-1 in the mitochondria, thereby resulting in accumulation of MCL-1.

## Stabilized MCL-1 by LANA inhibits caspase-3-mediated apoptosis

As MCL-1 plays a pivotal role in caspase-3-mediated apoptosis [5, 6], we next determined the impact of LANA-mediated stabilization of MCL-1 on the cellular apoptotic response. Three stable TREx/BJAB cell lines expressing empty vector, LANA, or LANA-P1 were treated with Doxy in the presence or absence of etoposide. Apoptosis was then examined by flow cytometry after propidium iodide (PI) and Annexin-V staining, or by immunoblotting with a caspase-3 specific antibody. As shown in **Fig 4A**, etoposide induced apoptosis of TREx/BJAB-vector (63.1 ± 3.48%) and TREx/BJAB-Au-LANA-P1 cells (67.4 ± 3.17%), but had little effect on TREx/BJAB-Au-LANA cells (21.9 ± 3.98%). Additionally, we observed that cleaved caspase-3 levels were greatly diminished in LANA-expressing cells than compared to vector and LANA-P1-expressing cells (**Fig 4B**). It has been reported that etoposide destabilizes MCL-1 by targeting FBW7 in a GSK-3β-mediated phosphorylation-dependent manner [33,34]. Consistent with these reports, we found that etoposide decreased endogenous MCL-1 level markedly in vector- and LANA-P1-expressing cells, whereas LANA protected MCL-1 from etoposide-induced degradation (**Fig 4B**). This result further supports the data presented in **Fig 3B** showing that LANA reduced ubiquitination of MCL-1 via interaction with FBW7. Moreover, endogenous FBW7 IP assay showed that LANA efficiently competes with FBW7 for MCL-1 interaction both in the presence and absence of etoposide (**Fig 4C**). Taken together, these results demonstrate that LANA inhibits caspase-3-mediated apoptosis by increasing the stability of MCL-1 in a FBW7-dependent manner.

## MCL-1 is highly accumulated upon KSHV infection via LANA-FBW7 interaction-dependent manner

In order to examine the effect of LANA-mediated stabilization of MCL-1 in the context of the KSHV infection, we first generated a LANA-P1 mutant KSHV by replacing Theronine at amino acid 177 in LANA encoded in KSHV BAC16 to Alanine (rKSHV-BAC16-LANA-P1) via "scarless" mutagenesis [35]. To rule out the possibility of second-site mutations, we also constructed a revertant clone in which the wild-type (WT) LANA sequence was restored (rKSHV-BAC16-Rev) (**Fig 5A**). After validating the recombinant constructs by *Nhe I* restriction enzyme digestion and DNA sequencing (**Fig 5A**), we produced infectious virus using iSLK cell lines carrying WT, LANA-P1, and Rev KSHV BAC16 clones (S2A Fig) [35]. We then determined the effect of LANA-P1 mutant on the viral gene expression as well as production of infectious virus. To this end, we induced lytic reactivation of KSHV in iSLK cells, harboring

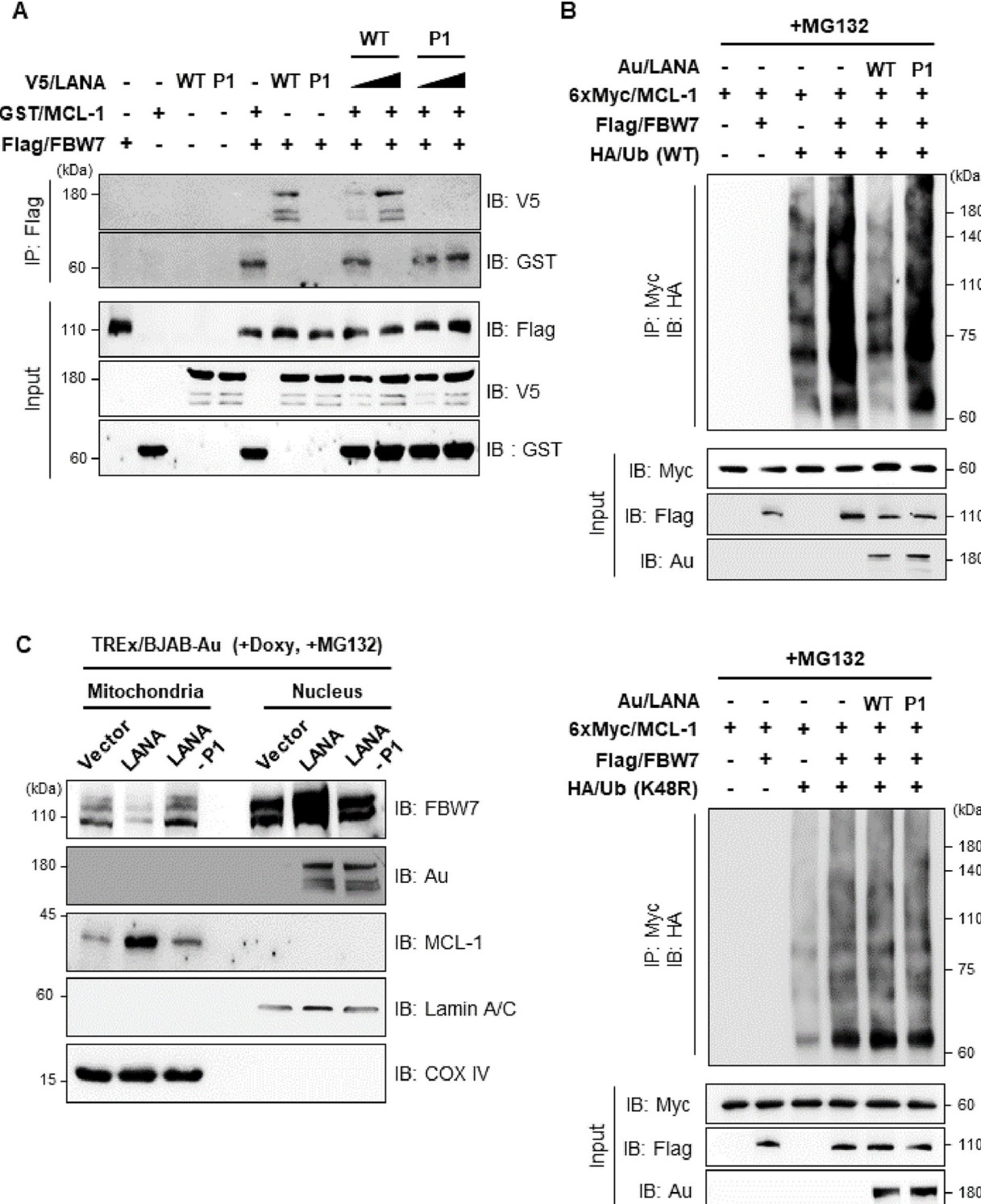

**Fig 3. The KSHV LANA-FBW7 interaction dramatically reduces ubiquitination of MCL-1.** (A) 293T cells were transfected with different combinations of plasmids. The cells were then treated with CHX (20 μg/ml) for the indicated periods, followed by IP with an anti-Flag antibody and IB with either an anti-V5 or an anti-GST antibody. (B) 293T cells were co-transfected with the indicated LANA constructs along with the FBW7, MCL-1, and ubiquitin constructs. The cells were then treated with MG132 (10 μM), followed by IP with an anti-Myc antibody and IB with an anti-HA antibody. (C) TREx/BJAB cells stably expressing vector, LANA-WT or LANA-P1 were treated with Doxy (1 μg/ml) together with MG132 (10 μM), followed by subcellular fractionation and IB with indicated antibody. Nuclear membrane protein Lamin A/C and mitochondrial protein COX-IV were shown as an indicator of nuclear or mitochondrial fraction, respectively.

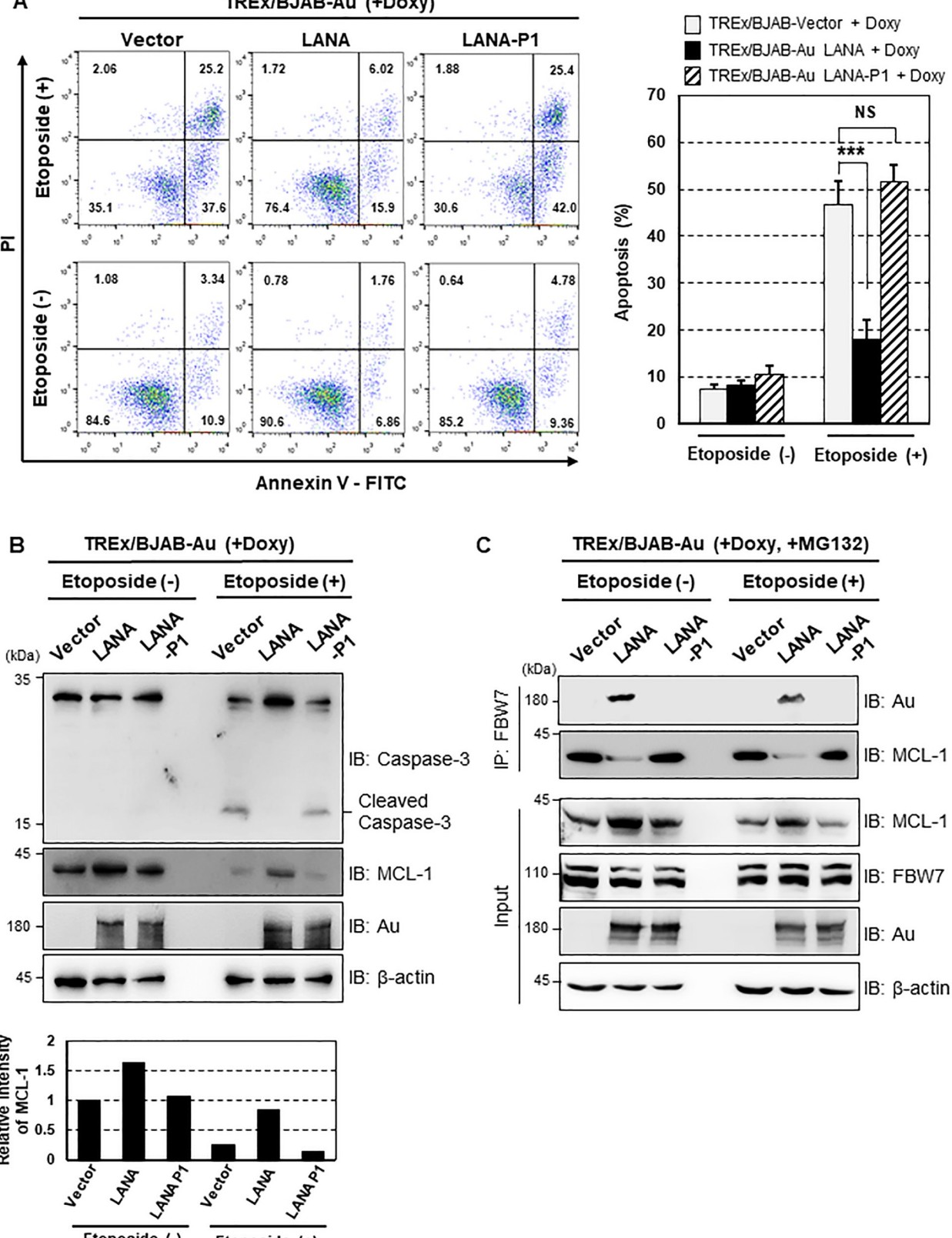

**Fig 4. KSHV LANA-mediated MCL-1 stabilization inhibits caspase-3-dependent apoptosis.** (A, left) TREx/BJAB cells stably expressing LANA-WT or LANA-P1 were stimulated with 1 µg/ml of Doxy, treated with etoposide (50 µM) for 24 h, and then assessed for Annexin V and propidium iodide (PI) staining by FACS analysis. Numbers indicate the percentage of cells in each quadrant. (A, right) The cell populations representing early (lower right quadrant) and late (upper right quadrant) apoptosis were summarized and shown as a graph. *P*-values were calculated using Student's *t*-test (compared with the results for the T/BJAB-vector cell line). NS (Non-significant), $p > 0.05$; ***, $p < 0.0005$; N = 4. (B) Upon stimulation with Doxy (1 µg/ml), cells were also treated with etoposide (25 µM) for 24 h before harvesting. Cell lysates were then used for IB with an anti-caspase-3 antibody. (C) Cells were harvested after treatment with Doxy (1 µg/ml) and etoposide (25 µM) for 24 h, followed by treatment with MG132 (10 µM) for 6 h. Cell lysate were then used for IP with anti-FBW7 antibody and IB with either anti-Au or anti-MCL-1 antibodies.

rKSHV-BAC16-LANA-P1, rKSHV-BAC16-Rev, and rKSHV-BAC16, and measured both virus production and the expression of the immediate-early (RTA), early (ORF6, ORF45, K2), and late (K8.1) viral proteins. We found that LANA-P1 mutant KSHV produce comparable amount of virus compared to WT KSHV (S2B Fig). Accordingly, the expression levels of viral proteins tested did not appear to be affected by LANA-P1 mutant either (S2C Fig), suggesting that LANA-P1 mutant does not affect virus production and viral gene expression. To examine whether LANA also has the ability to induce MCL-1 stabilization in KSHV-infected cells, we established BJAB cell lines with rKSHV-BAC16, rKSHV-BAC16-LANA-P1, and rKSHV-BAC16-Rev (S2D Fig). We found that MCL-1 is highly accumulated in both BJAB-rKSHV-BAC16 and BJAB-rKSHV-BAC16-Rev cells, but not in BJAB-rKSHV-BAC16-LANA-P1 (**Fig 5B**). In addition, we observed that MCL-1 stabilized via rKSHV-infection markedly increased cells proliferation (**Fig 5C**), and dramatically reduced apoptosis measured by PI staining (**Fig 5D**). Collectively, our results demonstrate that KSHV LANA appears to be a critical viral protein required for MCL-1 stabilization during KSHV infection.

## LANA-mediated stabilization of MCL-1 is essential for survival of KSHV-associated PEL cells

Since LANA, a master regulator of KSHV latency, is highly expressed in all latently infected tumor cells, we examined the relative expression levels of MCL-1 in KSHV-positive PEL cell lines compared to a KSHV-negative cell line. Strikingly, we detected higher expression of MCL-1 in KSHV[+] BCBL-1, BC-3 cells, and KSHV[+]EBV[+] BC-1 cells than in KSHV[−] BJAB cells (**Fig 6A**). Consistent with this, endogenous MCL-1 ubiquitination was significantly lower in BCBL-1 cells than in BJAB cells upon MG132 treatment, indicating that high levels of MCL-1 protein correlates inversely with ubiquitination of endogenous MCL-1 (**Fig 6B**). In addition, we further verified that LANA competes with MCL-1 for FBW7 binding via endogenous FBW7 IP in BCBL-1 cells (S3 Fig). Furthermore, siRNA-mediated depletion of MCL-1 in KSHV[+] PEL cell lines, such as BCBL-1, BC-3 and BC-1, but not in KSHV[-] BJAB cells, increased cell death, suggesting that KSHV[+] PEL cell lines require MCL-1 for survival (**Figs 6C** and S4A). Thus, elevated expression of MCL-1 in PEL cells prompted us to assess the therapeutic potential of MCL-1 inhibitors AT-101 and A-1210477. AT-101 is a pan-BCL-2 inhibitor with anti-MCL-1 activity [36,37], while A-1210477 is a selective MCL-1 small-molecule inhibitor [37,38]. Treatment of BJAB and PEL cells with AT-101 or A-1210477 suppressed cell growth and induced cell death in KSHV[+] PEL cells but had very marginal effect on BJAB cells (**Figs 6D,** S4B, S4C and S4D). Taken together, these data indicate that LANA-mediated stabilization of MCL-1 via sequestering FBW7 from MCL-1 is essential for the survival of PEL cells.

## MCL-1 promotes tumorigenesis of KSHV[+] BCBL-1 cells

Next, to evaluate the anti-tumor activity of MCL-1 inhibitors in KSHV[+] BCBL-1 cells *ex vivo*, we performed soft agar colony assays in the absence or presence of MCL-1 inhibitors (AT-101

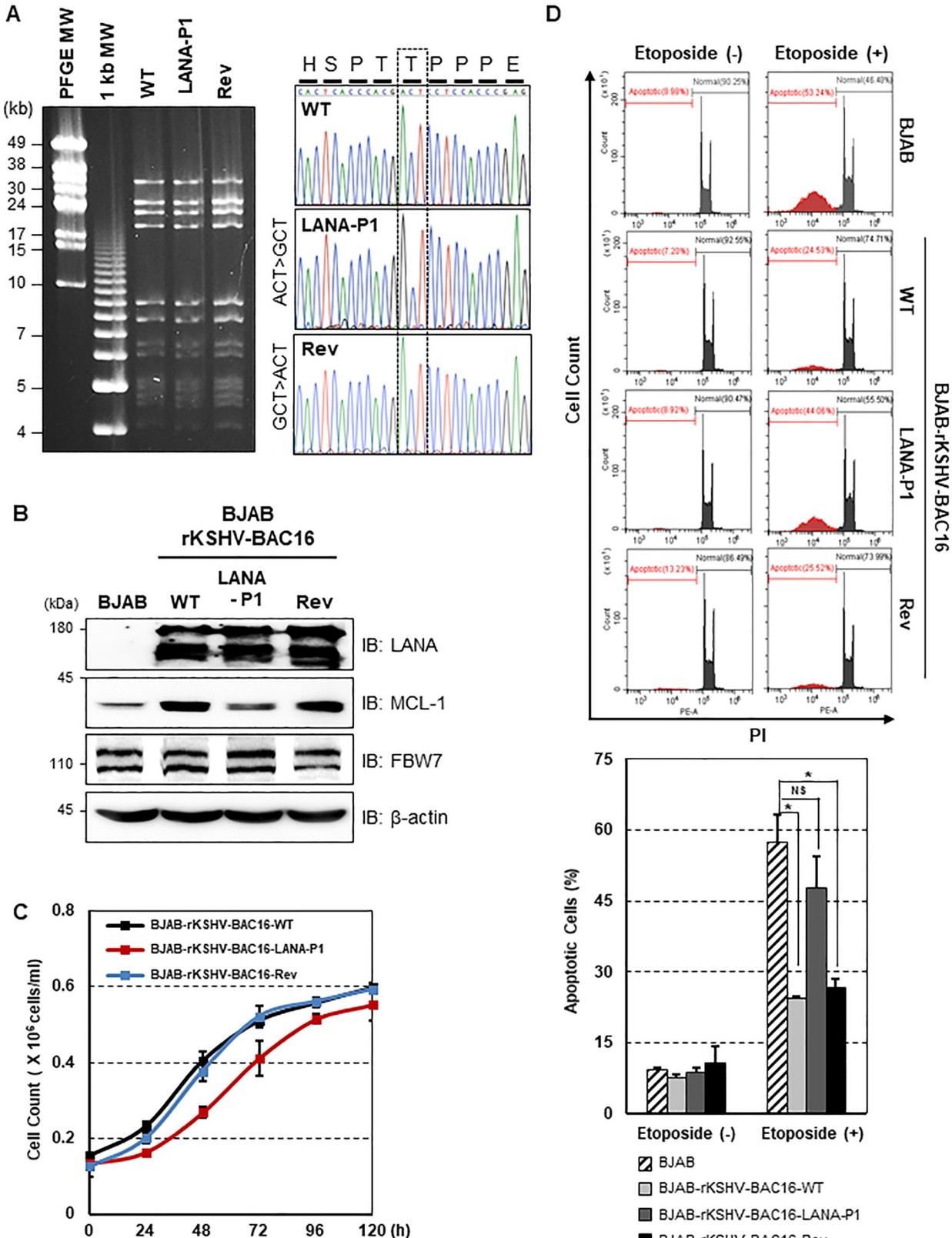

**Fig 5. KSHV-infection induces the MCL-1 stabilization in BJAB cells.** (A) (Left panel) BAC DNAs were digested with *NheI* restriction enzyme and subjected to PFGE analysis. (Right panel) BAC16 clones were confirmed by Sanger DNA sequencing. (B) WT, LANA-P1, Rev recombinant

KSHV-infected BJAB were harvested and equal amounts of cell lysates were used for IB with the indicated antibodies. (C) BJAB-rKSHV-BAC16-WT, BJAB-rKSHV-BAC16-LANA-P1, and BJAB-rKSHV-BAC16-Rev cells were counted every 12 h. Error bars represent the SEM for three independent experiments. (D) After treatment with etoposide (50 μM), Cells were then stained with PI and carried out the FACS analysis. Data represent the mean ± SEM and *P*-values were calculated by Student's *t*-test. NS, $p > 0.05$; *, $p < 0.05$; N = 3.

or A-1210477), followed by crystal violet staining to count colonies. Notably, both co-treatment of BCBL-1 cells with MCL-1 inhibitors and treatment of cells with MCL-1 inhibitors after colony formation led to marked inhibition of colony formation (**Fig 7** upper panel and S5A Fig upper panel), whereas BJAB cells successively formed colonies in soft agar in the presence or absence of MCL-1 inhibitors (**Fig 7** bottom panel and S5A Fig bottom panel). To confirm the direct effect of MCL-1 on KSHV-associated transformation, we depleted MCL-1 from BCBL-1 and BJAB cells using siRNA. As expected, depletion of MCL-1 considerably reduced the efficacy of colony formation by BCBL-1 cells, but not by BJAB cells (S5B Fig). Collectively, these results suggest that MCL-1 plays a crucial role in the tumorigenic properties of BCBL-1 cells.

## MCL-1 is a potential candidate drug target in KSHV⁺ PEL

To further evaluate the *in vivo* anti-tumor activity of MCL-1 inhibitors for PEL, we utilized NOD/SCID xenograft mice injected intraperitoneally with BCBL-1 cells, as described previously [39,40]. Two days after injection, mice began receiving a weekly intraperitoneal injection of either DMSO as a delivery vehicle or AT-101 lasting for 6 weeks. We observed abdominal distention in DMSO-treated mice but not in AT-101-treated mice (**Fig 8A**). Given that abdominal distention reflects the development of ascites, we collected and measured the volume of ascites produced in each mouse. On average, 1.77 ml ascites were collected from DMSO-treated mice versus 0.12 ml from AT-101-treated mice (**Fig 8B**). The volume of ascites correlated with the increase in body weight (**Fig 8C**), verifying that weight gain was attributable to tumor establishment. Previous studies reported that mice injected with KSHV⁺ PEL cells, including BCBL-1 cells, exhibited notable splenomegaly when compared with normal NOD/SCID mice due to infiltration of anaplastic cells [41]. Notably, the size of the spleen in AT-101-treated mice was obviously reduced compared to the DMSO-treated control mice (**Fig 8D**). KSHV⁺ PEL cells, including BCBL-1 cells isolated originally from patients typically expressed CD45 and CD38, but are usually negative for B-cell markers such as CD19, CD20, CD79, and PAX5 [42–44]. We also verified the expression of both CD45 and CD38 on the surface of ascites cells of our xenograft model by using FACS analysis, validating our *in vivo* system for studying tumorigenesis by PEL cells (S6 Fig). These results conclusively confirmed the anti-tumor activity of the MCL-1 inhibitor, AT-101, thereby identifying MCL-1 as a potential therapeutic target for treatment of KSHV⁺ PEL.

## Discussion

KSHV-associated PEL has an unfavorable prognosis and treatment options for the patients are highly limited [45]. This calls for an urgent need to understand its mechanism of progression to establish rational for developing alternative target-based therapeutics. Very recently, an attempt has been made at using genome-wide loss-of-function CRISPR screens to verify *MCL-1* as a critical host gene in PEL [10]. Moreover, tissue microarray technology revealed high expression of MCL-1 in tissue specimens from KSHV-associated malignancies [10,11]. Nevertheless, the molecular mechanism of increased levels of MCL-1 in PEL and its function in KSHV-associated pathogenesis have yet to be understood. Here, we showed that KSHV LANA interacts specifically with FBW7, which competes for MCL-1 binding and reduces

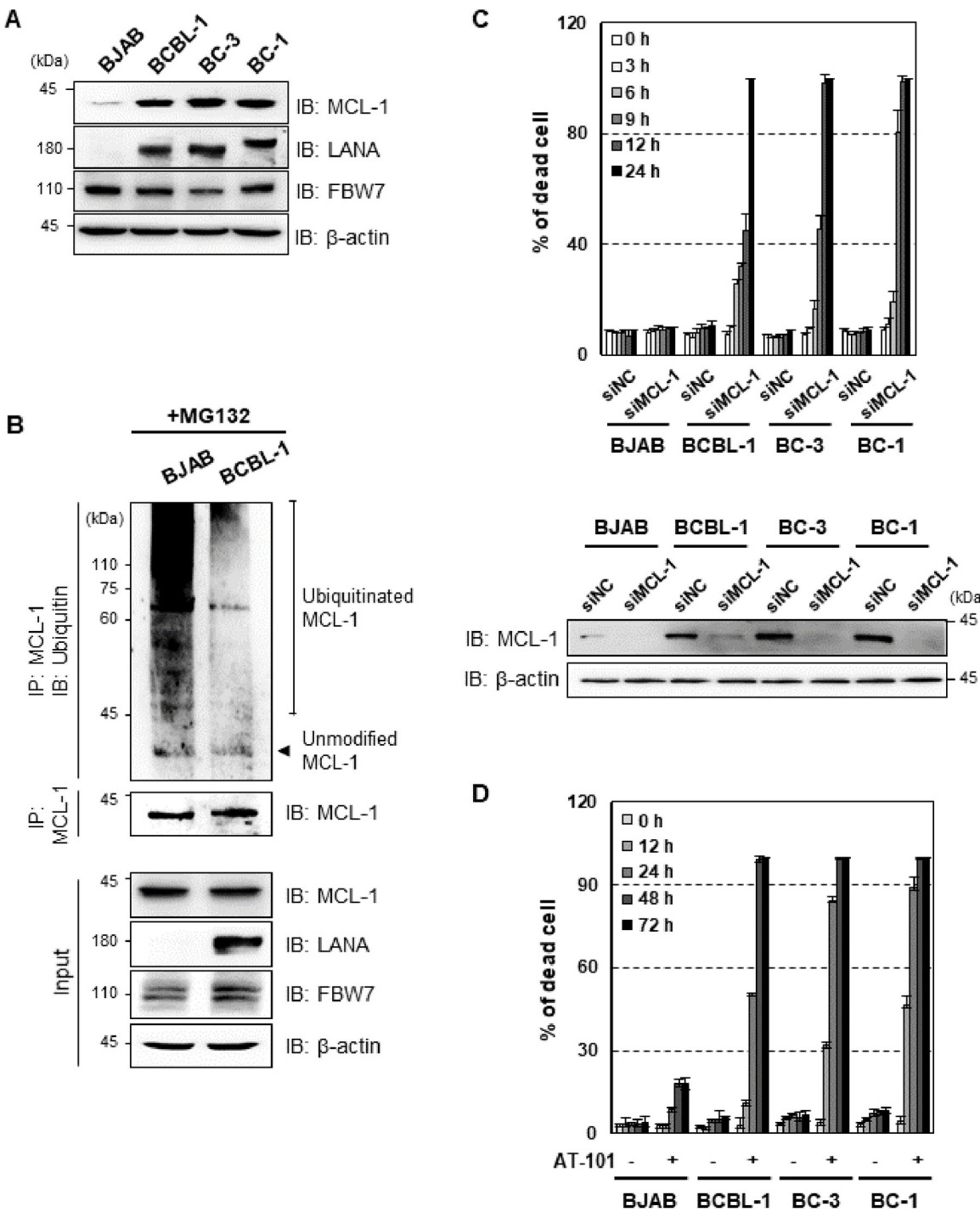

**Fig 6. MCL-1 is essential for PEL cell survival.** (A) KSHV⁻ BJAB cell, KSHV⁺ PEL cell (BCBL-1 and BC-3), and KSHV⁺EBV⁺ PEL cell (BC-1) lysates were subjected to IB with anti-LANA and anti-MCL-1 antibodies. β-actin is shown as a loading control. (B) The cells were treated with MG132 (10 μM), followed by IP with an anti-MCL-1 antibody and IB with an anti-ubiquitin antibody. (C) BJAB and KSHV-infected PEL cell lines were transfected with negative control siRNA or siRNA that target MCL-1 (100 pmol) for the indicated periods of time, followed by staining with trypan blue solution. The effect on MCL-1 protein was determined by counting of trypan blue-stained cells. Cells were transfected with scrambled or MCL-1 siRNA for 12 h, followed by immunoblotting (IB) with anti-MCL-1 or anti-β antibodies. (D) Cells were treated with 10 μM AT-101 (MCL-1 inhibitor) for the indicated periods of time and then stained with trypan blue solution.

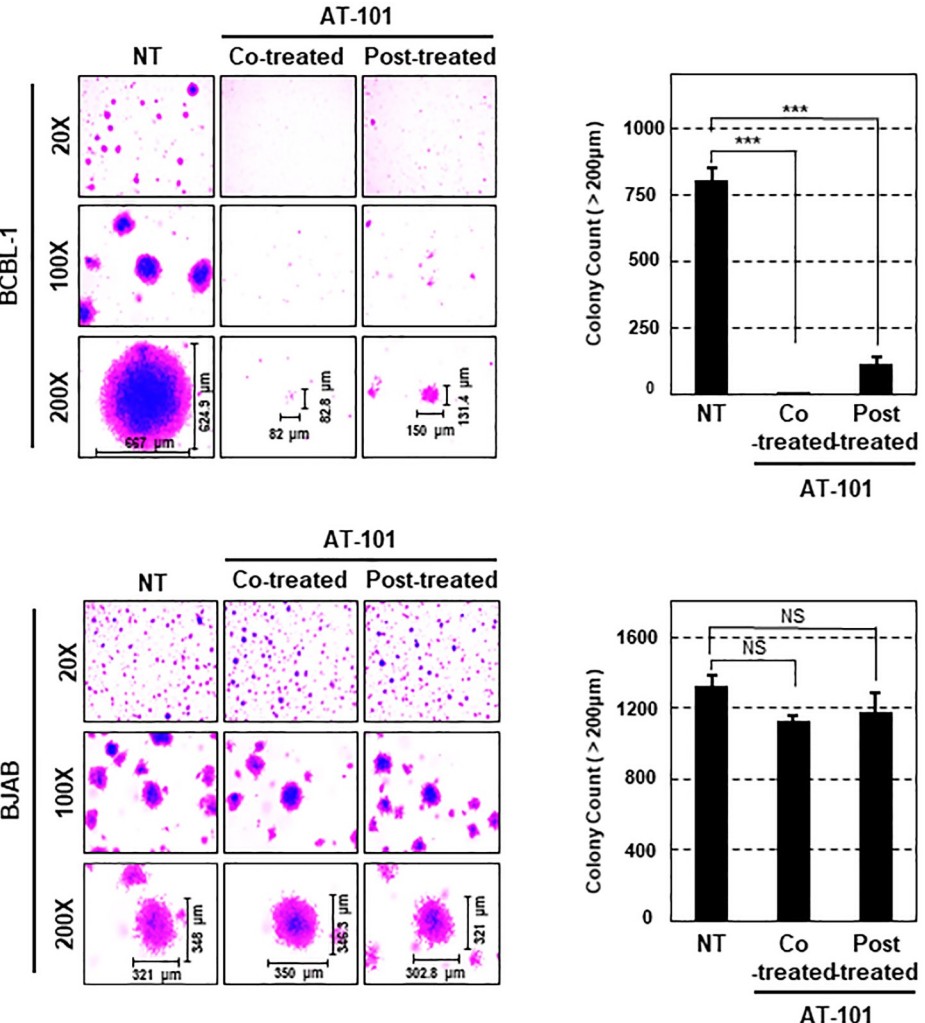

**Fig 7. MCL-1 is required for colony formation by PEL cells.** BJAB and BCBL-1 cells were plated in soft agar with or without AT-101 (10 μM) for 2 weeks. Representative pictures were captured at the indicated magnifications and the number of colonies larger than 200 μm was counted. Data represent the mean ± SEM. Statistical analysis was conducted using a two-tailed Student's $t$-test. NS, $p > 0.05$; ***, $p < 0.0005$.

ubiquitination of MCL-1, thereby increasing its stability. Furthermore, siRNA-mediated inhibition of MCL-1 and small-molecule inhibitors (AT-101 and A-1210477) effectively suppressed the anti-apoptotic function of MCL-1, eventually promoting PEL cell death. More importantly, targeting MCL-1 using AT-101 inhibited PEL development in xenograft tumor models, suggesting that MCL-1 inhibitors may serve as effective novel anti-cancer therapeutics for PEL.

Alternative splicing of human FBW7 gene produces three isoforms (α, β, and γ), which show different subcellular localizations given that FBW7 harbors both a nuclear localization signal (NLS) and a nuclear export signal (NES) [46,47]. FBW7-α, which is mainly localized to the nucleus, is thought to carry out most of FBW7 functions even though specific actions for other isoforms have also been described [47]. Recently, Song *et. al.*, reported that FBW7 translocates from the nucleus to cytoplasm when cells are infected with VSV to stabilize RIG-I [48]. Moreover, phosphorylation of FBW7 at specific residues can cause the exclusion of FBW7

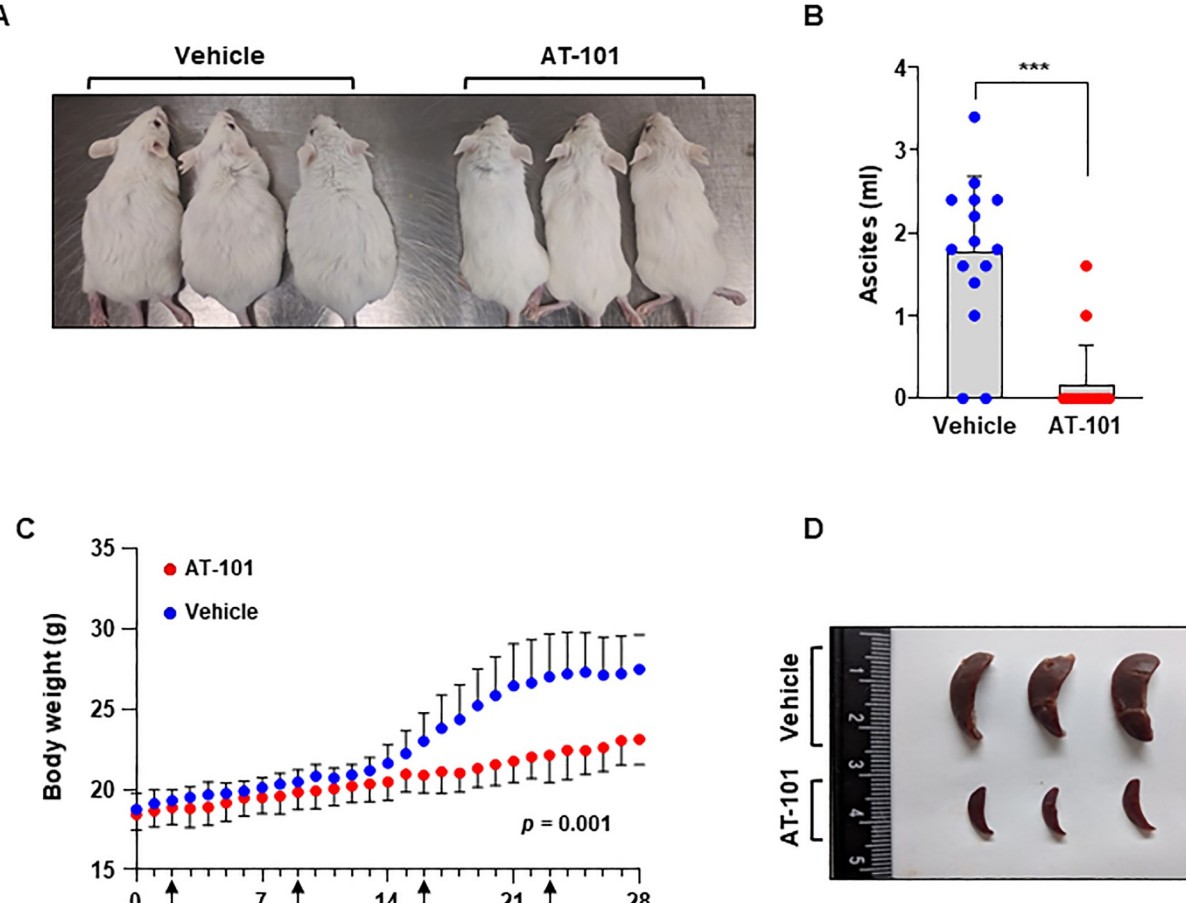

**Fig 8. MCL-1 inhibitor treatment prevents KSHV⁺ BCBL-1-mediated tumor formation in NOD/SCID mice.** (A) BCBL-1 cells were injected intraperitoneally into NOD/SCID mice, followed by treatment with AT-101 once a week. Mice were euthanized using $CO_2$ at 6 weeks after cell injection. Animals treated with AT-101 did not develop the distention observed in vehicle-treated animals. (B) After 28 days, mice were sacrificed and the volume of ascites fluid was measured. The number of dots in ascites volume represents the number of sacrificed mice. Error bars represent the SEM for two independent experiments. ***, $p < 0.0005$. (C) Body weight of DMSO or AT101 treated mice was recorded daily. $p = 0.001$ (D) The spleens in AT-101-treated animals were smaller than those in vehicle-treated animals.

from the nucleus resulting in its mislocalization, which can impede the role of FBW7 in degradation of specific substrates [47,49,50]. Since most of the FBW7 substrates are proto-oncogenes, relocalization of FBW7 is also likely to derive transformation. In accordance with this, our study shows that the interaction of LANA with FBW7 elevated the presence of FBW7 in the nucleus while decreased FBW7 level in the mitochondria, which correlated with increased protein expression of MCL-1. Thus, our results suggest that LANA-mediated mislocalization of FBW7 leads to the stabilization of MCL-1.

Additional studies will be necessary to understand the mechanism by which LANA-FBW7 interaction can specifically alter MCL-1 protein expression while not affecting the expression of other FBW7's substrates. There is one possibility that may explain how LANA specifically sequesters FBW7 from MCL-1. FBW7 has been shown to bind to conserved CPD motifs in FBW7's substrates, which have different binding affinities depending on what specific residues are phosphorylated in CPD. High-binding affinity CPDs include two phosphorylated residues (both "0" position and "+4" position), while low-affinity CPDs contain a negatively charged residue in the second site ("+4" position, such as glutamic acid (E)) [4,51,52]. As shown in **Fig**

**1A**, LANA contains a negatively charged residue at the second site within the decoy CPD motif, suggesting that LANA has a low affinity CPD that acts as a weak binding substrate of FBW7. Strikingly, the majority of FBW7's substrates have two phosphorylation sites in their CPD motifs except MCL-1 [2,4,53,54]. Thus, we hypothesize that both LANA and MCL-1 having a low-affinity CPD, LANA can compete with MCL-1 for FBW7 binding but not with proteins that have high-affinity CPD (e.g. c-Myc), which can bind stronger to FBW7. Additional studies will be required to test this hypothesis.

Although LANA-FBW7 interaction did not dysregulate the expression of other FBW7's substrates, it is noteworthy that LANA employed other strategies to upregulate them. In the case of c-Myc, LANA interacts with glycogen synthase kinase 3 beta (GSK-3β), which interferes with the phosphorylation of c-Myc that can block the binding of FBW7 to its CPD motif, leading to the stabilization of c-Myc [29]. Indeed, mounting data reported that GSK-3β phosphorylates the central threonine ("0" position) of the CPD in substrates containing a priming site (like c-Myc, cyclin E, and Jun), resulting in timely degradation of several substrates of FBW7. These results suggest that LANA has evolved to dysregulate the tumor suppressor function of FBW7-related signaling pathway by targeting either FBW7 or its upstream factor, like GSK-3β. Numerous studies have shown that LANA has various properties that could clearly contribute to KSHV-mediated tumorigenesis, including inhibiting G1-cell cycle arrest [28], stimulating S-phase entry via upregulation of cyclin D1 and β-catenin [55], or suppressing p53-mediated apoptosis [56]. These results indicate that the KSHV LANA specifically targets either protein-protein interactions or post-translational modifications, like ubiquitination, which can contribute to KSHV-associated pathogenesis, further emphasizing the important role of LANA as a viral oncogenic factor.

In summary, our studies provide experimental evidence for the role of LANA in governing MCL-1-mediated intrinsic apoptotic pathways via regulation of one UPS component, FBW7, within the FBW7-MCL-1 axis to contribute to KSHV pathogenesis. Notably, MCL-1 plays an oncogenic role in regulating the survival and tumorigenicity of PEL cells but not in Burkitt's lymphoma cells (e.g., BJAB). We also supply a mechanistic explanation for why MCL-1 is enriched in PEL, as well as in Kaposi's sarcoma (KS). These findings indicate that high expression of MCL-1 in PEL cells supports its oncogenic role, making it a potential drug target. In particular, we demonstrate for the first time that inhibiting MCL-1 function markedly represses PEL-induced tumor growth in xenograft models. Taken together, these results increase our understanding of how KSHV utilizes the cellular ubiquitin system to promote KSHV-associated pathogenesis.

## Materials and methods

### Ethics statements

Animal experiments were approved by the Seoul National University Institutional Animal Care (SNU-190506-5-2) and Use Committee and performed following its guideline.

### Cell culture and cell line construction

293T and Vero cells were cultured in Dulbecco's modified Eagle's medium supplemented with 10% fetal bovine serum (FBS) and 1% penicillin-streptomycin (P/S) (Corning). Most suspension cells (BCBL-1, BC-1, BC3, and BJAB) and tetracycline-inducible TREx/BJAB cells [57] were maintained in RPMI 1640 medium supplemented with 1% P/S and 10% FBS or 10% Tet system approved FBS (TaKaRa), respectively. Transient transfection was performed using PEI (Sigma) or FuGENE HD (Promega) according to the manufacturers' instructions. To establish BJAB cells expressing wild-type (WT) or mutant LANA in a tetracycline-inducible manner, pcDNA/FRT/

To-LANA/Au or pcDNA/FRT/To-LANA-P1/Au was transfected together with the Flp recombinase expression vector pOG44. Twenty-four hours after transfection, cells were selected using 200 μg/ml of hygromycin B (Invitrogen). The detailed procedure was delineated previously [58].

## Chemicals and antibodies

Chemicals were purchased from the following manufacturers: cycloheximide (CHX), doxycycline (Doxy), sodium butyrate (NaB), MG132, and etoposide were obtained from Sigma; A-1210477 was obtained from Cayman Chemical; and AT-101 was procured from Apex Bio. Primary antibodies were purchased from the following manufacturers: anti-MCL-1 (4572), anti-ubiquitin (3933), anti-mTOR (7959), and anti-KSHV LANA (LNA-1; 4013) were purchased from Abcam; anti-FBW7 (A301-720A) was obtained from Bethyl; anti-MCL-1 (sc-12756), anti-GST (sc-138), anti-Lamin A/C (sc-625), anti-ORF45 (sc-53883), and anti-K8.1 (sc-65446) were obtained from Santa Cruz; anti-β-actin (A-5441), anti-FLAG (F1804), and anti-tubulin (GTU-88) were purchased from Sigma; anti-V5 (MA5-15253) was obtained from Invitrogen; anti-Au (901901) was purchased from BioLegend; anti-HA (M180-3) was procured from MBL; anti-Cyclin E (2983) and anti-COXIV (4850) were purchased from cell signaling technology; anti-K2 (13-214-050) was obtained from ABI. Myc antibody was kindly provided by Dr. Jin-Hyun Ahn (Sungkyunkwan University). Anti-RTA and anti-ORF6 antibodies were generously provided by Dr. Yoshihiro Izumiya (University of California, Davis) and Dr. Gary Hayward (Johns Hopkins University), respectively.

## Plasmid construction

pcDNA-Flag-FBW2, pcDNA-Flag-FBW5, pcDNA-Flag-FBW7, and pcDNA-Flag-βTrcP were provided by Dr. Jae U. Jung (Lerner Research Institute of Cleveland Clinic, Cleveland, OH, USA). The coding sequence of MCL-1 was amplified from template DNA (purchased by Addgene) via PCR analysis and subcloned into the pENTRY vector at the restriction sites *Eco*RI and *Bam*HI. Subsequently, expression plasmids for 6xMyc-MCL-1 or GST-MCL-1 were generated on a pDEST background using the Gateway technology (Invitrogen). LANA-P1, LANA-P2/3, and LANA-P1/2/3 were generated using a site-directed mutagenesis kit (Agilent) according to the manufacturer's instructions. The primers used for these LANA mutant plasmids were as follows: P1 forward-CTCACCCACGCTGCTCCACCCGA; P1 reverse-TCGGG TGGAGCAGCCGTGGGTGAG, P2/3 forward-ACACTAAACCCAATATGTCAGGCGGCC CCAGTCGCTCCCCCTAGATGTGACTTCGCC; P2/3 reverse-GGCGAAGTCACATCTAG GGGGAGCGACTGGGGCCGCCTGACATATTGGGTTTAGTGT.

## Generation of recombinant KSHV clones

The LANA mutant (P1) KSHV and its revertant were generated by bacterial artificial chromosome (BAC)-based homologous recombination using the KSHV clone BAC16 in the *E. coli* strain GS1783, as previously described [35, 59]. The recombinant BAC16 clones were verified by Sanger sequencing and *Nhe I* restriction enzyme digestion followed by pulsed-field gel electrophoresis analysis. Primers used for making LANA mutant P1, in which at amino acid 177 Threonine was changed to Alanine, are the following: LANA T177A bacF: 5' CTCCACACTC ACCCACG**GC**TCCTCCACCCGAGCCTCCCTCCAAGTCGTCACCAGACTCTTAGGATG ACGACGATAAGTAGGG, LANA bacR: 5' ACGCAGGGTAGACGGAGCTAAAGAGTCT GGTGACGACTTGGAGGGAGGCTCGGGTGGAGGAACCAATTAACCAATTCTGATTA G. The primer used for making the revertant (Rev) from the P1 mutant, in which at amino acid 177 Alanine was changed back to Threonine, along with LANA bacR is the following:

LANA A177T bacF: 5' CTCCACACTCACCCACG**ACT**CCTCCACCCGAGCCTCCCTCCA AGTCGTCACCAGACTCTTAGGATGACGACGATAAGTAGGG.

## Establishment and characterization of iSLK-BAC16 cell lines

The iSLK-BAC16 WT, LANA-P1, and Rev cell lines were established as previously described (Golas G, Virology, 2020). Briefly, iSLK cells were transfected with BAC16 DNA using FuGENE HD (Promega). Two days after transfection, we started the selection of iSLK cells carrying BAC16 using 1 mg/ml hygromycin B. The GFP$^+$/hygromycin B resistant iSLK-BAC16 cell lines were established after 3 weeks of hygromycin B selection. For virus reactivation, the cells were treated with 1 μg/ml Doxy and 1 mM NaB. Cells were harvested for immunoblot analysis upon 72 hours after chemical treatment. At 5 days post-induction, the cell culture supernatant was collected, filtered through 0.45 μm SFCA syringe filter and purified for virion-associated DNA. Supernatant was treated with proteinase K followed by phenol-chloroform extraction of virion-associated DNA, which was then measured by KSHV DNA specific real-time quantitative PCR. Based on a standard curve using serial dilution of BAC16 DNA, we calculated the amount of virus produced by iSLK-BAC16 cell lines.

## Generation of rKSHV-BAC16-infected BJAB cells

The iSLK-BAC16-WT, iSLK-BAC16-LANA-P1, or iSLK-BAC16-Rev cell lines were reactivated by 1 mM NaB together with 1 μg/ml Doxy for 3 days. Subsequently, BJAB cells were cocultured with three different reactivated iSLK-BAC16 cell lines. After 4 days, infected BJAB cells were separated from the adherent cell lines by gently shaking the tissue culture plates and transferred into the new plate. These collected cells were treated with 200 μg/ml hygromycin B. After three weeks of culture, polyclonal BJAB cells infected with rKSHV-BAC16 were designated as BJAB-rKSHV-BAC16 cells.

## Subcellular fractionation assay

Cells were lysed in fractionation buffer (20 mM HEPES [pH 7.4], 10 mM KCl, 2 mM MgCl$_2$, 1 mM EDTA, 1 mM EGTA, 1 mM PMSF) and then passed through a 27 G needle (30x). The nuclear pellet was obtained after centrifugation at 720 g for 15 min. The supernatant was centrifuged at 12000 g for 15 min. The obtained mitochondria pellet was lysis in the buffer (150 mM NaCl, 0.5% NP40, 50 mM Tris [pH 8.0]). Both nucleus and mitochondria samples were mixed with SDS loading dye and were analyzed by immunoblotting.

## siRNA

Negative-control siRNA and siRNA for MCL-1 (AM51331-#120644) were purchased from Ambion. All siRNAs were used at a final concentration of 100 pmol and transfected into cells using the Neon transfection system according to the manufacturer's instructions (Invitrogen).

## Ubiquitination assay

BJAB and BCBL-1 cells transfected with different combination of plasmids were collected followed by treatment with 10 μM MG132 for 6 h. Cells were lysed in lysis buffer (150 mM NaCl, 0.5% NP40, 50 mM Tris [pH8.0], 0.1% Triton X-100, 50 mM NaF, 5 mM Sodium pyrophosphate) supplemented with protease inhibitors (Complete Mini, Thermofisher), incubated at 95˚C, and sonicated at 4˚C for 12 s with 1–3 s pulses using a Digital Sonifier (Branson). After immunoprecipitation with an anti-Myc antibody, immunoprecipitates were analyzed via immunoblotting.

## Immunofluorescence assay

Cells were fixed with 4% paraformaldehyde (Sigma) and permeabilized with 0.1% Triton X-100 (Sigma). All detailed processes were described previously [59].

## Apoptosis assay

TREx/BJAB cells (5 x $10^5$ cells/ml) stably expressing WT or mutant LANA were cultured for 24 h and then treated with 50 μM etoposide for up to 24 h. The detailed procedure was previously described [58].

## Soft agar foci formation assay

BCBL-1 (2.5 x $10^5$ cells/ml) or BJAB (2.0 x $10^5$ cells/ml) cells were seeded in agar matrix in six-well plates. After solidification, medium containing 10 μM MCL-1 inhibitor was added on top of the cell/agar matrix. After 2 weeks, colonies were stained with 0.05% crystal violet (Daejung) for 30 min and washed. The colonies were viewed using the EVOS-5000 imaging system (Invitrogen).

## Xenograft models

BCBL-1 cells were washed three times with PBS and injected intraperitoneally into 6-week-old female NOD/SCID mice (Koatech, Gyunggi-do, Korea). Each group utilized 15 mice. AT-101 dissolved in DMSO was diluted in PBS and intraperitoneally administered to mice at a dose of 2.5 mg kg$^{-1}$ once per week after 2 days from BCBL-1 cell inoculation. Inhibition of tumor growth was monitored daily for 28 days by comparing body weight between AT-101- and vehicle-treated mice. After euthanasia, peritoneal ascites was collected using a 5-ml syringe, and the spleen and solid tumor in the peritoneum were isolated. All data were analyze by GraphPad Prism 8.3 software and two-tailed unpaired t-test.

## Analysis of ascites cells

To isolate cells from ascites, ascites were filtered using a 70-μm cell strainer (BD Falcon) and washed three times with ice-cold PBS. Ascites and BCBL-1 cells were incubated with anti-mouse CD16/32 (Clone 2.4G2, BD Pharmingen 553142) antibody and then stained with anti-human CD45 (Clone HI30, BD Pharmingen 555483) and anti-human CD38 (Clone HIT2, eBioscience 47-0389-42) antibodies or isotype controls. After washing three times with ice-cold PBS containing 1% BSA (MP Biomedicals 9048-46-8), cells were analyzed using a Cyto-FLEX S Flow Cytometer (Beckman Coulter) and FlowJo v10.6.0 software (Tree Star Inc).

## Statistics

Statistical analyses were performed using GraphPad Prism software. *P*-values were calculated by Student's *t*-test and $p < 0.05$ denoted statistical significance. Data are presented as the mean ± SD.

## Supporting information

**S1 Fig. Localization patterns between wild-type LANA and different mutant LANA.** Vero cells transiently expressing LANA wild-type (WT) or three different LANA phospho-dead mutant plasmids were fixed and viewed by confocal microscopy using anti-Au (green) representing LANA. Nuclei (blue) were stained with DAPI.
(TIF)

**S2 Fig. Characterization of LANA mutant BAC16.** A. Three different iSLK-BAC16 cell lines expressed GFP. B. These iSLK-BAC16 cells were treated with 1 μg/ml of Doxycyline and 1 mM sodium butyrate for 72 hours. After reactivation, virus production was measured at 5 days post-induction based on virion-associated DNA purified from cell culture supernatant. The *p*-values were calculated applying Student's two-tailed t-Test. C. The same set of cells were analyzed by immunoblotting using indicated antibodies. Representative IE (RTA), E (ORF6, ORF45, K2), and L (K8.1) viral proteins are shown along with LANA and the loading control tubulin. The average of six biological replicates is shown. D. BJAB cells were co-clutured with reactivated iSLK cells harboring BAC16 KSHV WT, LANA-P1, and Rev. These cells were treated with 200 μg/ml hygromycin B. After establishing BJAB-BAC16 cell lines, cells were monitored for GFP expression in KSHV-BAC16 by fluorescence microscopy.
(TIF)

**S3 Fig. Competing LANA with MCL-1 for endogenous FBW7 interaction.** Both BJAB and BCBL-1 cells were treated with MG132 (10 μM) followed by immunoprecipitation with anti-FBW7 antibody and immunoblotting with either an anti-LANA or an anti-MCL-1 antibody.
(TIF)

**S4 Fig. Crucial role of MCL-1 in PEL cell survival.** A. BJAB and KSHV-infected PEL cell lines were transfected with negative control siRNA or siRNA that target MCL-1 (100 pmol) for the indicated periods of time, followed by staining with trypan blue solution. Either negative control siRNA- or MCL-1 siRNA-transfected cells were counted for the indicated periods of time. B. AT-101-treated cells were stained with trypan blue solution, followed by counting for the indicated periods of time. Data represent the mean ± SD of the combined results from three independent experiments. C and D. Cells were treated with 10 μM of A-1210477 for the indicated time periods. C, Staining with trypan blue solution; D, Live cells counting. Data represents the means (± SD) of the combined results from three independent experiments.
(TIF)

**S5 Fig. Critical function of MCL-1 in colony formation in KSHV⁺ PEL cells.** A. BCBL-1 and BJAB cells were plated in soft agar with or without 10 μM of A-1210477 for two weeks. Total colonies larger than 200 μm in representative pictures captured at indicated magnifications were counted. The data represent the means ± SEM. Statistical analysis was performed using two-tailed Student's *t*-test. Non significance (NS), $p > 0.05$; *, $p < 0.05$; ***, $p < 0.0005$. B. Negative control siRNA- or siMCL-1-transfected BJAB and BCBL-1 cells were plated in soft agar for 2 weeks (upper). ****, $p < 0.00005$. Negative control siRNA or siMCL-1-transfected cells were subjected to IB with anti-MCL-1 and anti-β-Actin antibodies (bottom).
(TIF)

**S6 Fig. Verification of expression of PEL-surface markers in ascites cells of KSHV⁺PEL xenograft models.** The BCBL-1 cells and cells from ascites fractions were stained for PEL-surface markers CD45 and CD38, and then subjected to FACS analysis. Isotype controls, PE mouse IgG1, κ and mouse IgG1, κ antibodies were used as a negative control for FACS analysis.
(TIF)

## Acknowledgments

We thank Dr. Jae U Jung and Dr. Jin Hyun Ahn for providing reagents.

## Author Contributions

**Conceptualization:** Hye-Ra Lee.

**Data curation:** Yeong Jun Kim, Hye-Ra Lee.

**Formal analysis:** Yeong Jun Kim, Chan Woo Kim, Hye-Ra Lee.

**Funding acquisition:** Zsolt Toth, Nam Hyuk Cho, Hye-Ra Lee.

**Investigation:** Yeong Jun Kim, Yuri Kim, Abhishek Kumar, Chan Woo Kim.

**Methodology:** Yeong Jun Kim, Yuri Kim, Zsolt Toth.

**Project administration:** Nam Hyuk Cho, Hye-Ra Lee.

**Resources:** Hye-Ra Lee.

**Supervision:** Hye-Ra Lee.

**Validation:** Hye-Ra Lee.

**Visualization:** Yeong Jun Kim, Yuri Kim, Hye-Ra Lee.

**Writing – original draft:** Hye-Ra Lee.

**Writing – review & editing:** Zsolt Toth, Hye-Ra Lee.

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
