## [Decision Letter · Decision Letter 0]

4 Aug 2020

Dear Hey-Ra,

Thank you very much for submitting your manuscript "Kaposi’s Sarcoma-associated Herpesvirus Latency-associated Nuclear Antigen Dysregulates Expression of MCL-1 by Targeting FBW7" for consideration at PLOS Pathogens. As with all papers reviewed by the journal, your manuscript was reviewed by members of the editorial board and by several independent reviewers. In light of the reviews (below this email), we would like to invite the resubmission of a significantly-revised version that takes into account the reviewers' comments.

We cannot make any decision about publication until we have seen the revised manuscript and your response to the reviewers' comments. Your revised manuscript is also likely to be sent to reviewers for further evaluation.

Sincerely,

Pinghui Feng

Associate Editor

PLOS Pathogens

Erik Flemington

Section Editor

PLOS Pathogens

Kasturi Haldar

Editor-in-Chief

PLOS Pathogens

orcid.org/0000-0001-5065-158X

Michael Malim

Editor-in-Chief

PLOS Pathogens

orcid.org/0000-0002-7699-2064

Reviewer's Responses to Questions

**Part I - Summary**

Reviewer #1: Myeloid cell leukemia 1 (MCL-1) is an anti-apoptotic protein that is critical for survival of immune cells including T lymphocytes, B lymphocytes, and macrophages. Recently, MCL-1 has emerged also as an essential factor for the survival of KSHV-infected primary effusion lymphoma (PEL) cell lines. Here, the paper re-invited the essential role of MCL-1 in KSHV infection using MCL-1 depletion or inhibitors in cultured PEL cell lines and a xenograft mouse model of PEL. As an original part of the manuscript, the authors surveyed the molecular mechanism by which MCL-1 is stabilized in the PEL cell lines and proposes that LANA competes with MCL-1 for binding to the ubiquitin E3 ligase FBW7, thus reducing ubiquitination and degradation of MCL-1. This proposed model is supported by the use of the LANA mutant P1, identified by the authors, which is incapable of binding to FBW7. While the novelty of the paper is mainly represented by the molecular mechanism of action for LANA-mediated MCL-1 stabilization, key pieces of evidence are still missing, in particular, in the context of virus infection, and the results of polyubiquitination of MCL-1 is not convincing. In addition, the paper lacks necessary discussion on already published LANA interaction with FBW7 and on how the mitochondrial protein MCL-1 is regulated by the nuclear proteins LANA and FBW7.

Reviewer #2: In this interesting manuscript, Kim et al. describe a novel mechanism of how KSHV LANA stabilises MCL-1, an anti-apoptotic cellular protein, which has previously been shown by others to be highly expressed in and to be essential for the survival of primary effusion lymphoma (PEL) cells and to be a 'druggable' target to treat PEL in preclinical models. The manuscript by Kim et al. therefore contributes two important aspects to the KSHV literature: (i) it shows that KSHV LANA stabilises MCL-1 by binding to and thereby diverting the E3 ligase FBW7, which normally downregulates MCL-1; (ii) it describes two further MCL-1 or FBW-7 inhibitors that potently inhibit PEL growth in an in vivo preclinical model and thereby extends a previous report on another MCL-1 inhibitor, S63845, which had been shown by others to markedly inhibit the growth of of PEL cells in tissue culture. The experiments described in this very well written manuscript are of high technical quality.

**Part II – Major Issues: Key Experiments Required for Acceptance**

Reviewer #1: 1. LANA-mediated MCL-1 stabilization and related mechanisms were proposed mainly based on the studies using sole expression of LANA in KSHV-negative BJAB cell line and 293T cells. So, it is not appreciated how important LANA, in particular, via sequestering FBW7, contributes to the elevated expression of MCL-1 in KSHV-infected PEL cell lines as shown in Figure 5A. To address this point, it is desirable to use recombinant KSHV viruses, e.g., BAC16, harboring LANA wild-type (WT), knockout, and revertant with WT or the P1 mutant, for MCL-1 assays in BJAB cells. If applicable, it’d be also nice to identify and use a competitive inhibitory peptide that can block the interaction between LANA and FBW7 in KSHV-infected PEL cells.

2. The paper examined protein interactions using immunoprecipitation of ‘overexpressed’ proteins. Despite the presence of solid evidence, the interactions among LANA, FBW7, and MCL-1 should be further investigated in terms of biological relevance, e.g., i) following etoposide treatment in the BJAB cells expressing LANA WT or P1 and ii) in the PEL cell lines along with control cell BJAB. In either case, MG132 treatment may be needed to preserve MCL-1 protein to make the level of MCL-1 comparable as shown Figure 5B.

3. It is not convincing that the ubiquitinated bands shown in Figures 3B and 5B represent polyubiquitination of ‘MCL-1’. The paper did not describe how MCL-1-associated proteins, probably also ubiquitinated, were removed from the MCL-1-immunoprecipitated complex in the Materials and Methods section. Sonication could not remove MCL-1 interactions with other cellular proteins. In addition, in Figure 3B, controls are missing such as transfection of MCL-1 and HA-Ub or HA-Ub alone.

An increase on polyubiquitination of MCL-1 does not necessarily implicate enhanced proteasomal degradation of MCL-1. Thus, it is desirable to identify what type of polyubiquitination, e.g., K48 and K63, of MCL-1 in the Figures and to examine whether LANA can suppress K48-linked polyubiquitination, which is involved in proteasomal degradation, of MCL-1.

4. In fact, the interaction of LANA and FBW7 (also known as SEL10) was previously reported by Lan K, et al. in PNAS (2007), who demonstrated that the oncogenic factor intracellular activated notch (ICN) could be stabilized by LANA competition with FBW7. ICN is a reminiscence of MCL-1. However, both studies show the discrepant results in FBW7-binding regions in LANA. It needs to be addressed. It is curious why ICN was not involved in Figure 2B.

Reviewer #2: no major criticisms. In my opinion, no additional experiments are needed to support the conclusions drawn in this manuscript.

**Part III – Minor Issues: Editorial and Data Presentation Modifications**

Reviewer #1: 1. MCL-1 is mainly detected and functional in mitochondria. How can the nuclear proteins LANA and FBW7 (Figure 1F) stabilize the mitochondrial protein MCL-1?

2. It is desirable to show the levels of primary (target) proteins, in addition to binding proteins, immunoprecipitated in Figures 1B, 1C, 1D, 1E, 3A, 3B, and 5B.

3. In the top panels of Figures 3B and 5B, how do the authors know that the bottom bands are ‘unmodified MCL-1’? It is unlikely to detect the MCL-1 bands with anti-HA or ubiquitin antibody. Is it possible that the bands are the heavy chain of IgG used in immunoprecipitation?

4. In lines 228-229, the statement, “LANA enhanced endogenous MCL-1 protein levels (Fig. 4B)”, should be rephrased; for example, LANA protected MCL-1 from etoposide-induced degradation. However, ratios of the MCL-1 proteins before and after etoposide treatment look similar among vector, LANA, and LANA-P1 cells. Thus, it may be unlikely that LANA protected MCL-1 from etoposide-induced degradation. It’d be helpful to show the relative band intensities of MCL-1 of each etoposide-treated cell compared to no etoposide treatment.

5. It’d be nice to show the levels of FBW7 in BJAB and the PEL cell lines in Figure 5B.

6. How many mice were used in each group of the xenograft experiment? In Figure 7B, does the number of dots in ascites volume represent the number of sacrificed mice? Detailed description and statistic justification of the animal experiments should be addressed.

Reviewer #2: 1. Fig. 1F, text line 161: co-localisation of LANAwt and LANA P1 with FBW7: if LANA is transfected into cells in the absence of the KSHV genome, it shows a diffuse nuclear distribution which does not lend itself to co-localisation studies. The experiment shown in fig. 1F therefore only shows that the LANA P1 mutant localises to the nucleus, not that it 'co-localises' with FBW7 (line 161 in the text). I assume that the authors may only have wanted to make the point that the LANA P1 mutant has a nuclear localisation. They should therefore change the wording in the text on line 161.

2. Fig. 5C bottom panel: the WB for BJAB cells suggests that MCL-1 expression levels in BJAB are similar to those in the three PEL cell lines, in contrast to Fig. 5A, which indicates that MCL-1 levels are higher in PEL cells than in BJAB (as also shown on several occasions in the remainder of the manuscript). Presumably this is due to a longer exposure of the WB for BJAB in Fig. 5C. Comparable exposure times should be used for the WBs shown in figure 5C.

3. Fig. 5D, text line 256: the text states that AT101 induced cell death in PEL cells but not in BJAB; according to fig. 5D, there is a small increase in the number of dead cells also in BJAB cells. AT1210477 also reduces the growth rate of BJAB in fig. S2D, albeit not to the same extent as in PEL cells. The corresponding statements in the text should be rephrased.

4. line 306: "one of attempt" - better: "an attempt"

PLOS authors have the option to publish the peer review history of their article (what does this mean?). If published, this will include your full peer review and any attached files.

Reviewer #1: No

Reviewer #2: No
---

## [Decision Letter · Decision Letter 1]

22 Nov 2020

Dear Hey-Ra,

We are pleased to inform you that your manuscript 'Kaposi’s Sarcoma-associated Herpesvirus Latency-associated Nuclear Antigen Dysregulates Expression of MCL-1 by Targeting FBW7' has been provisionally accepted for publication in PLOS Pathogens. As you can see from reviewer 1, there are some minor points that you will need to address. Please revise accordingly. 

Best regards,

Pinghui Feng

Associate Editor

PLOS Pathogens

Erik Flemington

Section Editor

PLOS Pathogens

Kasturi Haldar

Editor-in-Chief

PLOS Pathogens

orcid.org/0000-0001-5065-158X

Michael Malim

Editor-in-Chief

PLOS Pathogens

orcid.org/0000-0002-7699-2064

Reviewer Comments (if any, and for reference):

Reviewer's Responses to Questions

**Part I - Summary**

Reviewer #1: Kim et al present a revised manuscript that is considerably strengthened in its demonstration of LANA regulation of MCL-1 in the context of virus-infected cells. The authors have enthusiastically performed ubiquitination assays in response to my comments.

Reviewer #2: Kim et al. describe a novel mechanism of how KSHV LANA stabilizes MCL-1, an anti-apoptotic cellular protein, which has previously been shown by others to be highly expressed in and to be essential for the survival of primary effusion lymphoma (PEL) cells and to be a 'druggable' target to treat PEL in preclinical models. The manuscript by Kim et al. therefore contributes two important aspects to the KSHV literature: (i) it shows that KSHV LANA stabilizes MCL-1 by binding to and thereby diverting the E3 ligase FBW7, which normally downregulates MCL-1; (ii) it describes two further MCL-1 or FBW-7 inhibitors that potently inhibit PEL growth in an in vivo preclinical model and thereby extends a previous report on another MCL- 1 inhibitor, S63845, which had been shown by others to markedly inhibit the growth of PEL cells in tissue culture.

In this revised version of their manuscript, the authors have carefully addressed all the reviewers' suggestions. The manuscript is very well written and the reported experiments are of high technical quality.

**Part II – Major Issues: Key Experiments Required for Acceptance**

Reviewer #1: None

Reviewer #2: No further experiments required

**Part III – Minor Issues: Editorial and Data Presentation Modifications**

Reviewer #1: The authors did not quite understand the concern (minor Q2) raised on co-immunoprecipitation (co-IP) assays. The comment was to show the amounts of primary (target) proteins immunoprecipitated in Figures 1B, 1C, 1D, 1E, 3A, 3B, and 5B. The level of “co-IPed protein” relies on the amount of protein targeted primarily for immunoprecipitation. Thus, in addition to the input (lysate), the levels of the immunoprecipitated primary proteins should be shown; for example, in Fig. 1E, immunoprecipitated Flag-FBW7 should be presented by immunoblotting with anti-Flag antibody.

Reviewer #2: no further minor points identified

PLOS authors have the option to publish the peer review history of their article (what does this mean?). If published, this will include your full peer review and any attached files.

Reviewer #1: No

Reviewer #2: **Yes: **Thomas F. Schulz

---

## [Editor Report · Acceptance letter]

16 Dec 2020

Dear Dr. Lee,

We are delighted to inform you that your manuscript, "Kaposi’s Sarcoma-associated Herpesvirus Latency-associated Nuclear Antigen Dysregulates Expression of MCL-1 by Targeting FBW7," has been formally accepted for publication in PLOS Pathogens.

Best regards,

Kasturi Haldar

Editor-in-Chief

PLOS Pathogens

orcid.org/0000-0001-5065-158X

Michael Malim

Editor-in-Chief

PLOS Pathogens

orcid.org/0000-0002-7699-2064